# Computational prediction of intracellular targets of wild-type or mutant vesicular stomatitis matrix protein

Matthew C. Morris[1], Thomas M. Russell[2], Cole A. Lyman[1], Wesley K. Wong[2], Gordon Broderick[1]*, Maureen C. Ferran[2]*

1 Center for Clinical Systems Biology, Rochester General Hospital, Rochester, NY, United States of America,
2 Thomas Gosnell School of Life Sciences, Rochester Institute of Technology, Rochester, NY, United States of America

* Gordonbroderick@rochesterregional.org (GB); mcfsbi@rit.edu (MCF)

**Data Availability Statement:** Transcriptomic data generated for this work has been uploaded to the NCBI under SRA accession number

## Abstract

The matrix (M) protein of vesicular stomatitis virus (VSV) has a complex role in infection and immune evasion, particularly with respect to suppression of Type I interferon (IFN). Viral strains bearing the wild-type (wt) M protein are able to suppress Type I IFN responses. We recently reported that the 22–25 strain of VSV encodes a wt M protein, however its sister plaque isolate, strain 22–20, carries a M[MD52G] mutation that perturbs the ability of the M protein to block NFκB, but not M-mediated inhibition of host transcription. Therefore, although NFκB is activated in 22–20 infected murine L929 cells infected, no IFN mRNA or protein is produced. To investigate the impact of the M[D52G] mutation on immune evasion by VSV, we used transcriptomic data from L929 cells infected with wt, 22–25, or 22–20 to define parameters in a family of executable logical models with the aim of discovering direct targets of viruses encoding a wt or mutant M protein. After several generations of pruning or fixing hypothetical regulatory interactions, we identified specific predicted targets of each strain. We predict that wt and 22–25 VSV both have direct inhibitory actions on key elements of the NFκB signaling pathway, while 22–20 fails to inhibit this pathway.

## 1 Introduction

Viral infection is recognized in the cytoplasm of infected cells by pathogen-associated molecular pattern receptors such as retinoic acid inducible gene-I (RIG-I). This leads to activation of transcription factors, including NFκB, IRF3, and IRF7, that induce antiviral cytokines such as TNF-α, IL-6, and Type I interferons (IFNα and IFNβ) [1–3]. RIG-I-dependent activation of IRF3 and consequent IFN gene expression has been well studied in VSV-infected cells [4–6]. However, VSV also induces NFκB-dependent expression of many other cytokines, whose activation and suppression have been less well examined.

Many viruses have evolved countermeasures that prevent the expression or function of the host antiviral response [2]. Vesicular stomatitis virus (VSV), the prototypical member of the Rhabdovirus family, has been widely studied as a model system to investigate the mechanisms

PRJNA670227. Data URL: https://www.ncbi.nlm.nih.gov/bioproject/PRJNA670227.

**Funding:** MCF R15CA246419 National Cancer Institute (National Institutes of Health) https://www.cancer.gov/ The funders had no role in study design, data collection and analysis, decision to publish, or preparation of the manuscript.

**Competing interests:** The authors have declared that no competing interests exist.

by which viruses evade the host's Type I interferon (IFNα and IFNβ) response. This is among the first antiviral defenses activated in an infected cell, where IFN proteins are secreted outside the cell and activate receptors on neighboring cells, triggering a signal transduction cascade that results in the induction of many antiviral genes whose products limit viral replication [1, 3].

Mutant strains of VSV are often associated with defective immune evasion functions, limiting their ability to infect cells with functioning IFN responses [7]. However, antiviral responses induced by IFNα and IFNβ are perturbed in many cancers, leaving them susceptible to infection by "oncolytic" viruses such as VSV [8–10]—even mutant strains without immune evasion mechanisms. Healthy normal cells are not harmed because they mount an antiviral response which blocks virus replication, while cancer cells usually have critical defects in their antiviral response pathways, rendering them susceptible to viral infection [11]. However, some cancers are resistant to VSV because their IFN response pathways remain intact [12–16]. Understanding these differences and the mechanisms by which VSV interacts with innate immunity is therefore essential to its continued development as a potential oncolytic agent [13].

A hallmark of wild-type (wt) VSV infection is the suppression of Type I IFN responses through one or more virus-encoded suppressors [17]. The most prominent of these is the matrix (M) protein, which is crucial for shutoff of host transcription [18–20], inhibition of nuclear-cytoplasmic transport of host RNAs [21–23], inhibition of host translation [24–27], and is sufficient to suppress IFN gene expression in the absence of other viral components [19, 28]. There is also a strong correlation between a virus's ability to inhibit host transcription and its ability to suppress IFN expression [28]. Wt VSV rapidly inhibits host RNA and protein synthesis and is a poor inducer of IFN [29]. In contrast, the mutant VSV strain T1026R1 (R1) [3] contains a single amino acid mutation at position 51 of the M protein [M(M51R)] [19], which abrogates its ability to inhibit host RNA and protein synthesis [30] and makes it a strong inducer of IFN [31, 32].

Since VSV is sensitive to the effects of IFN, it is plausible that the virus might utilize multiple mechanisms to evade the host's antiviral response. We recently found that the wt M protein alone was sufficient to inhibit virus-driven NFκB activation independently of infection, and that this inhibition was abrogated by the M(M51R) mutation. We also identified a mutant M protein–M(D52G)–in VSV isolate 22–20 and determined that this virus also activated NFκB [33]. Its sister plaque isolate, 22–25, did not contain a mutation in this highly conserved region of M and retained the ability to block NFκB activation [33]. IFN mRNA and protein were produced in L929 cells infected with viruses encoding the M(M51R) mutation, however little to no IFN mRNA or protein was produced in wt, 22–25, or 22-20-infected L929 cells [34]. It is thus likely that VSV M protein has multiple independent means of interfering with immune activation by acting on distinct cellular targets.

Understanding general mechanisms by which VSV regulates NFκB-dependent innate immune responses (IFN, apoptosis, autophagy) would be of great benefit for future identification of potential pharmacological targets to overcome VSV resistance in cancer cells with functioning IFN responses without permitting unwanted infection of healthy cells. In this work, we assembled a computational model of the RIG-I/NFκB pathways involved in host defense against VSV. This model was constrained to support transcriptomic data on the response of murine L929 fibroblasts to infection by wt, 22–20, and 22–25 VSV. Simulations of the model agreed on likely specific targets of different mutant strains of VSV, suggesting mechanistic explanations for the differential ability of these strains to block Type I IFN responses in host cells. Specifically, our simulations predict that wt and 22–25 VSV selectively inhibit NFκB-dependent signaling via IKKβ, while the 22–20 VSV strain, which bears the mutant M(D52G) protein, fails to inhibit this pathway.

## 2 Materials and methods

### 2.1 Cells, viruses, and infections

Murine L929 fibroblast monolayers (ATCC CCL-1) were grown in complete media containing Eagle's Minimum Essential Medium (EMEM) supplemented with 10% Horse Serum (HS). VSV field isolates 22–20 and 22–25 were generous gifts from Philip Marcus (University of Connecticut) and have been previously described [32, 35]. The heat resistant (HR) strain of the Indiana serotype of VSV was used as the wt virus [17]. All viruses were grown on either baby hamster kidney cells or Vero cells as previously described [3, 36]. L929 cells were infected with each virus at a multiplicity of infection (MOI) of 5 PFU/cell. Virus was adsorbed in EMEM for 1 h at 37°C in the absence of serum, after which complete medium was added.

### 2.2 RNA sequencing

Total RNA was isolated from infected L929 cells at 1 hour and 3 hours post infection (hpi) using the TRIZol Plus RNA Isolation Kit (Life Technologies) according to the manufacturer's directions. To ensure that the highest quality RNA is used in this analysis, 4 biological replicates of each condition were isolated. RNA was quantitated via nanodrop and bioanalyzer. Full workflow integrated service (RNA-Seq through Data QC and Analysis) was provided by ProteinCT Biotechnologies (Madison, WI). This workflow closely follows the pipeline outlined by Pertea et. al. [37] with modifications. cDNA libraries were prepared using the Illumina TruSeq strand specific mRNA sample preparation system (Illumina). Briefly, mRNA was extracted from total RNA using polyA selection, followed by RNA fragmentation and cDNA synthesis. The quality of the cDNA library was checked using the Agilent 4200 TapeStation. The libraries were sequenced (Single end 100bp reads) using the Illumina HiSeq4000. Ten samples per lane were run, with final counts reaching over 20 million reads per sample. The HISAT2 aligner software was used to map the raw data from the Illumina reads that was provided by ProteinCT to the GRCm38 genome (with annotations for snps and transcripts) [38]. SAMtools [39] was then used to convert the SAM output files from HISAT2 into BAM files. These converted files were subsequently compared to the Ensembl gene annotations (v90) GTF file using Stringtie [40]. The Stringtie output (estimated counts) were then converted to raw counts to use as input for differential expression analysis using R Bioconductor packages. PCA was performed as implemented in base R [41], and visualizations used the ggplot2 package [42].

### 2.3 Model assembly

Regulatory interactions (edges) between known components of the RIG-I and NFκB signal transduction pathways were identified by mining published scientific literature using the Pathway Studio platform (Elsevier Life Sciences, Amsterdam), which uses MedScan [43] natural language processing engine to infer the direction and effect (activation or inhibition) of regulatory actions. Each gene which showed significant dynamic variation over time or across viral infection by ANOVA was included in the search query as a network entity. Interactions between entities were limited to those with a documented mechanism of direct protein-protein interaction or modification (represented in the knowledge graph as "direct regulation", "protein modification", or "promoter binding" relations). The references supporting each interaction were validated by the authors to ensure that they were accurately interpreted. Gene expression for each network entity was expressed as a log2 fold change relative to the mock-infected cells. These fold changes were converted to discrete values by clustering onto gamma distributions using an expectation-maximization algorithm [44] implemented in MatLab (MathWorks). The activation of each entity in the network was constrained to the binary states of "active" (1) or "inactive" (0). Such binary logic

has been successfully employed in studies of intracellular signaling [45, 46]. Dynamic behavior was imparted to this regulatory circuit model using a logical formalism where the responses of each node under different combinations of input values are assigned using a truth table look-up of so-called K value entries [47, 48]. Other parameters calculated include the presence and effect of hypothetical edges from different VSV strains, as well as the activation threshold above which they exert their effects. Data from our own experiments (described above) were formulated as constraints which predictions based on any given set of model parameters must satisfy. Adherence to these constraints was expressed as a percentage of the maximum possible Manhattan distance between the predicted output and the reference experimental data (0–100%), with lower percentages representing closer adherence. Parameterization was conducted using a constraint satisfaction-based optimization method implemented under our group's BioModelChecker [49] suite of tools for the reverse engineering of biological networks.

## 2.4 Edge fixation

Model parameterization was conducted over multiple generations to determine the likely inclusion and polarity of the hypothetical connections between different VSV strains and each of the host molecular entities in the network circuit. After each generation, candidate models supporting the reference trajectories to within 5% departure were selected as described above. The adjusted polarity ($P_{adj}$) for each hypothetical edge was then calculated as $E_p/S-E_n/S$, where $E_p$ = the number of candidate models where the edge was included with positive polarity, $E_n$ = the number of solutions where the edge was included with negative polarity, and $S$ = the total number of solutions. Edges with $|P_{adj}|{\geq}0.5$ had a consistent inclusion and polarity in the majority of candidate models, and were therefore accepted into the circuit model and applied with their consensus polarity in the next generation of parameterizations, whereas edges with $|P_{adj}|{\leq}0.05$ were removed.

## 2.5 RT-PCR

L929 cells were challenged with 22–20 or 22–25 VSV at 5 for 6 hours. At the conclusion of this period cells were lysed and RNA harvested as previously described [6]. Briefly, total RNA was isolated from cells, reverse transcribed into cDNA, and the commercially available mouse Taq-Man expression assay (Mm00437121, Applied Biosystems) was used for Real-Time PCR analysis of TNFAIP3 mRNA production. Samples were run in triplicate and the HPRT endogenous control Taqman Gene Expression Assay (Mm00446968) was used for relative quantification. All calculations were done using the $2^{-ddCT}$ method. Statistical comparisons were performed on the log2 fold change by Student's t test in R.

# 3 Results

## 3.1 Data processing and network assembly

Transcriptomic data was first filtered for significantly differentially expressed genes by a one-way ANOVA searching for significant variation due to Virus (Mock-infected, 22–20, 22–25, and WT) and/or hours post-infection (HPI) (q<0.05 after Benjamini-Hochberg adjustment). A total of 4905 genes were found to be significantly affected by at least one of these variables. PCA performed on the significantly variable genes was able to explain some 88% of the total variance with the top 2 components. With respect to these 2 components, all 3 viruses strongly diverge from the mock samples by 3 HPI, though the 22–20 virus did not appear to induce substantial transcriptional activity by 1 HPI (Fig 1). The results of the PCA show substantial differences in the transcriptional response of L929 cells to each VSV strain, demonstrating the necessity for interrogation of the regulatory dynamics of these responses.

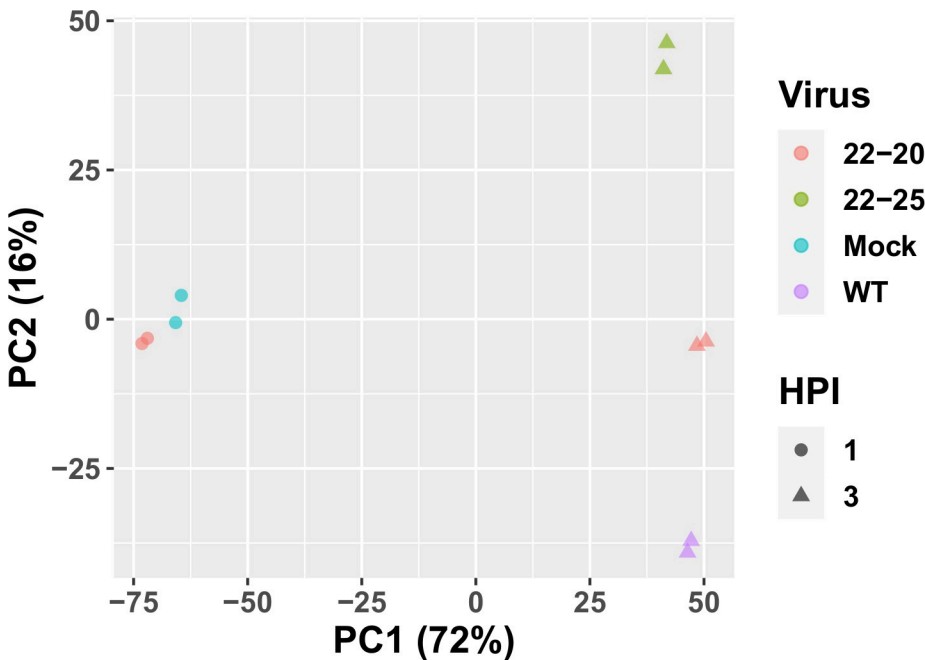

**Fig 1. PCA showing gene expression relative to mock-infected cells for variable genes.** Percentages are the fraction of total variance explained by each principal component.

Since these viral mutants all tended to significantly impact genes involved in immune response pathways, we focused our network modeling efforts on the subset of differentially regulated genes known to be members of RIG-I and NFκB pathways, which are highly influential in antiviral responses. Natural language processing of the peer-reviewed literature and mining of curated pathway databases yielded a network model incorporating 37 molecular entities involved in the RIG-I/NFκB signaling pathways governing Type I IFN responses (Fig 2). These entities were connected by 150 documented regulatory interactions ("edges"), supported by 8376 published references (median 18.5 per edge).

The mRNA expression of each network entity was expressed as the log2-transformed fold change relative to mock-infected cells. These fold changes were discretized by projection onto two gamma-distributions using expectation-maximization to compare the measured values with the different expected distributions under assumptions of relative activation or inhibition, resulting in values of 1 or 0 for each data point [44]. The median discrete value for each combination of virus strains and timepoints was then taken to represent the activation of each gene under each experimental condition. Trajectories were modeled as proceeding from the "mock" timepoint (State 1) with added virus through a 1h timepoint (State 2) before arriving at the 3h post-infection state (State 3). Part of the solution output is a prediction of the intervening events under unobserved experimental conditions. Input values were expressed in binary form, with entities not confidently assigned to either the maximum or minimum cluster value left unknown. Unknown values predicted for the 0h-timepoint in infected samples were constrained to be equivalent to one another.

### 3.2 Parameterization

The identification of regulatory logic parameters for the pathway network from discretized data was defined as a multi-objective optimization problem directed first to maximize

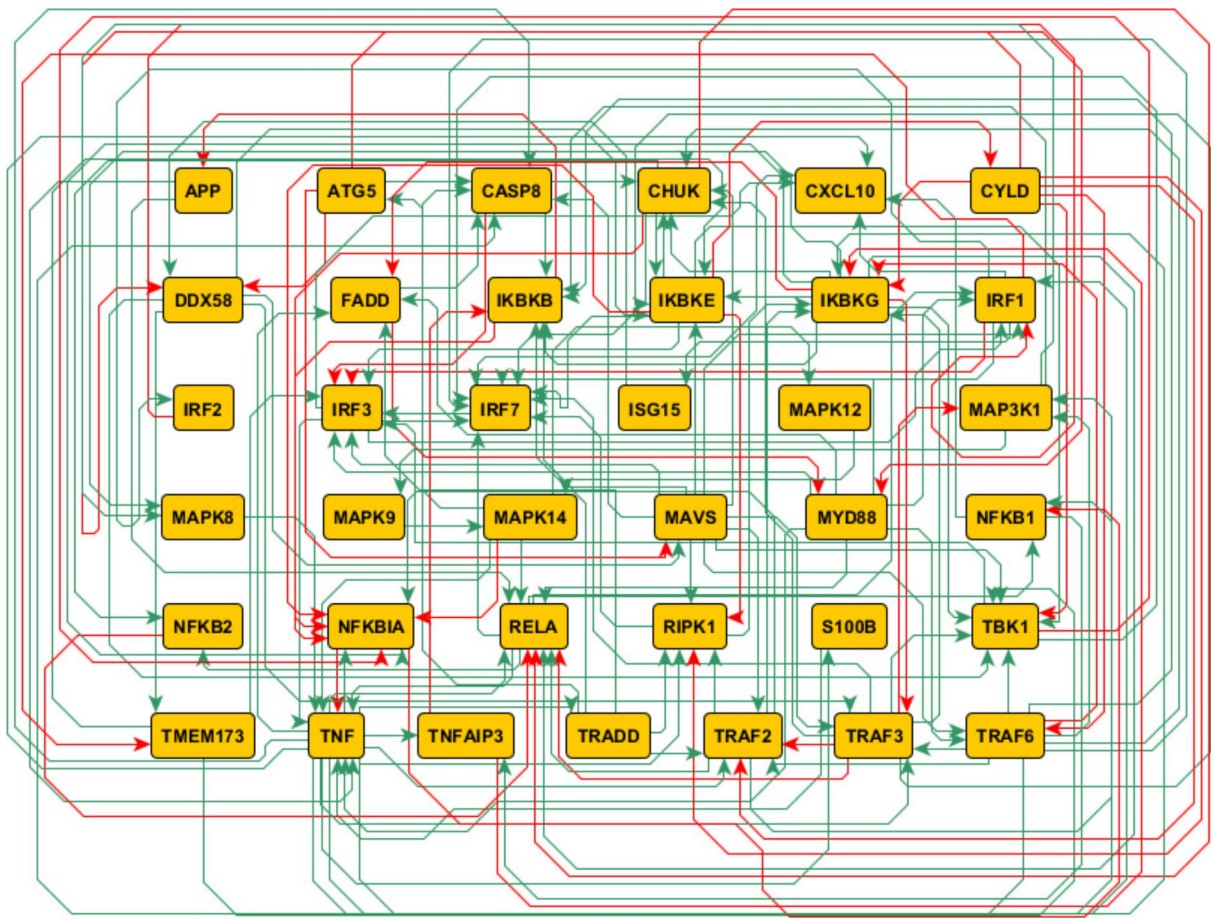

**Fig 2. Network circuit architecture based on mechanistic interactions documented in published scientific literature.** Green edges indicate activation of their targets and red edges indicate inhibition.

adherence to the input data and next to identify the most parsimonious models, i.e. the minimum number of targets required for 22–20, 22–25, and wt VSV to elicit the observed responses. We initially identified 145 candidate models adhering to the data to within 5% error (minimum error of 1.9%). Overall, 99.3% of logical K values were variable and did not display obvious clustering, suggesting a broad range of distinct competing regulatory kinetics each capable of reproducing the 3 sequentially observed expression profiles. The median pairwise distance between K matrices was 9555 out of a maximum 26870, indicating meaningful differences among the dynamic parameters (i.e. not limited to subtle changes without effect on output).

The median output trajectories from these models (Fig 3) closely follow the reference experimental data (Fig 4). For unobserved conditions, the predicted activation of network entities is determined by the dynamic parameters and connectivity associated with each candidate model. Thus, these models represent a family of mechanistic hypotheses consistent with the available data.

### 3.3 Edge fixation

A summary of the edges connecting each viral mutant to the network entities. Some edges have a high degree of both inclusion and consensus as to their effect. Using a simple majority

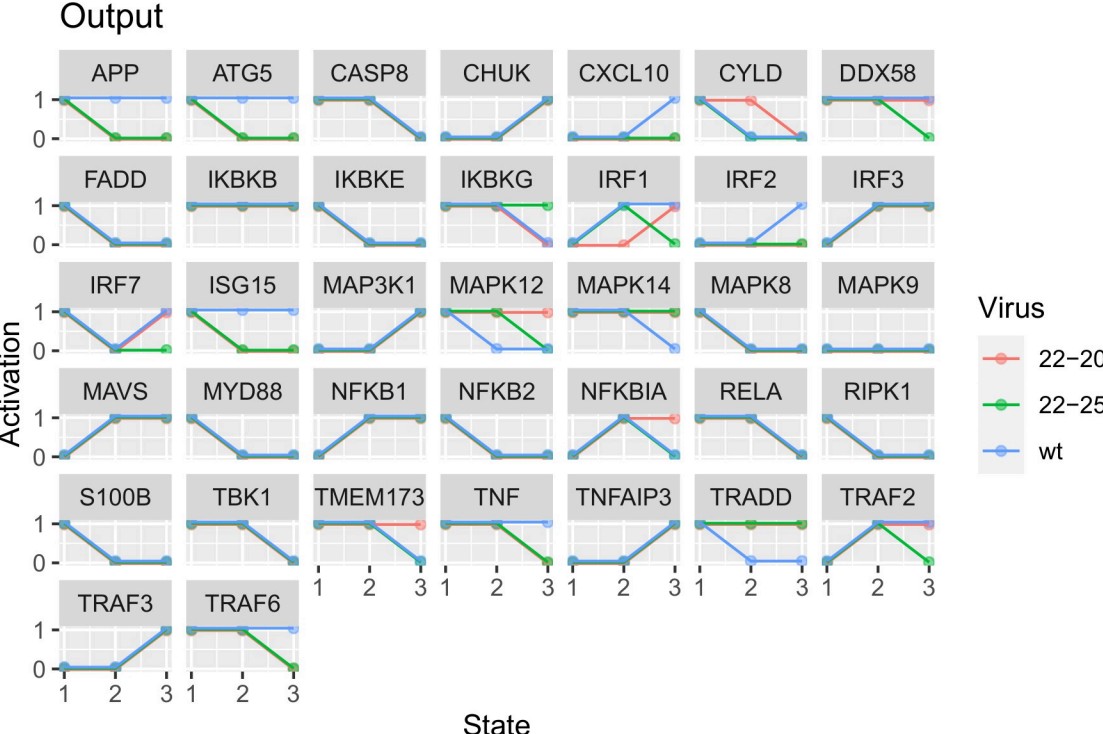

**Fig 3. Median output trajectories from all models which supported the available discretized reference trajectories to within 5% departure.** Timepoints missing from the initial reference trajectories were predicted according to the logical parameters of each candidate model.

voting scheme, edges with an adjusted polarity greater than or equal to 0.5 in the positive or negative direction were said to be "fixed" with relative confidence, while edges with an adjusted polarity less than or equal to 0.05 were considered here as unlikely to be necessary. Hypothetical edges confidently assigned after one generation of edge fixation were 22-20-TNFAIP3, 22-25-RELA, 22-25-TNFAIP3, and wt-TNFAIP3. The thresholds used for fixation are displayed in Fig 5.

After repeating parameterization with these additional constraints, we identified 41 distinct models capable of reproducing the data with <5% departure. The process of edge fixation was continued until succeeding generations failed to fix or remove any additional edges at the established confidence thresholds. In all, eight generations of edge fixation were required.

After the conclusion of this process, all but 4 of the hypothetical candidate edges were fixed with confidence, namely 22–20—NFKB1, 22–25—MAPK12, wt—ISG15, and wt—MAPK14 which remain indeterminate based on the available data. By the final generation of the model, IKBKB, MAPK8, MAVS, and TMEM173 were not direct targets of any of the viral strains under investigation. All three VSV strains were predicted to directly inhibit TNF and TRAF2, and to directly activate TRADD.

22–25 and wt VSV were predicted to have matching direct effects not shared by 22–20 on 8 targets (ATG5, CHUK, IKBKE, IRF1, NFKB1, RELA, RIPK1, and TNFAIP3), while 22–20 and wt had matching effects on 6 (APP, CXCL10, CYLD, FADD, IRF3, and MAPK9). 22–20 and 22–25 had matching effects on only 2 (MAPK14 and NFKB2).

Of note, 22–20 was the only virus to directly target IRF2, NFKBIA, and TRAF3, and 22–25 was the only virus to directly target DDX58, ISG15, S100B, and TBK1. There were no predicted targets unique to wt VSV.

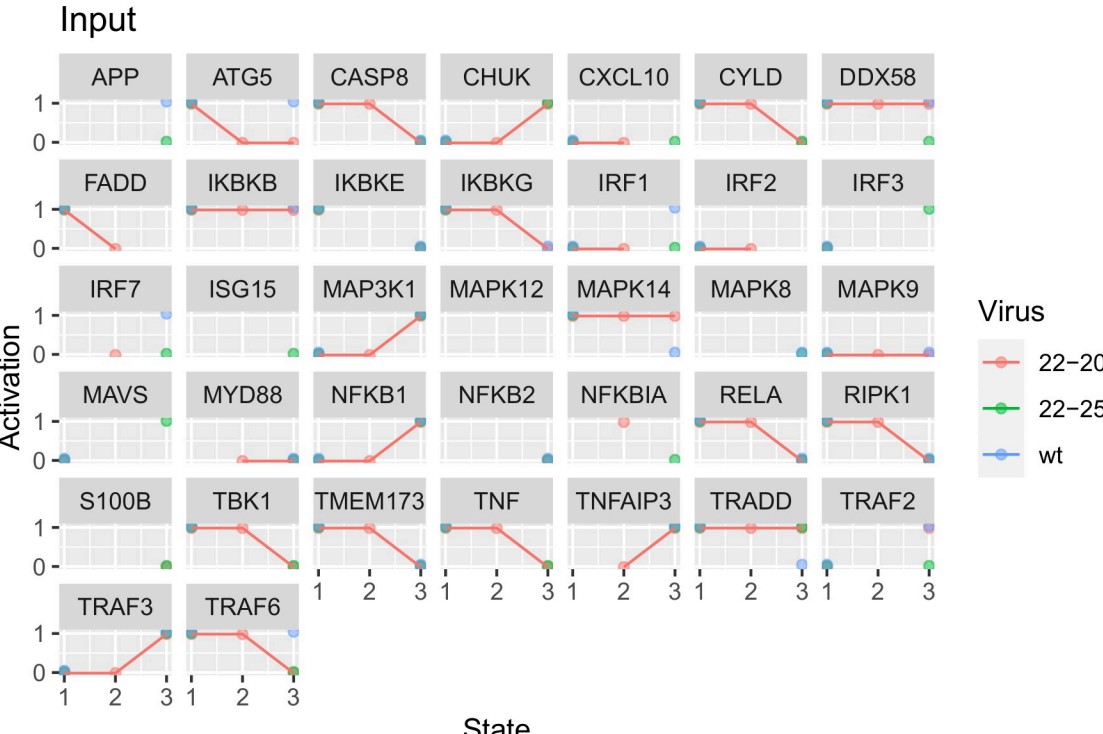

**Fig 4. Reference trajectories for parameterization showing the median discretized value for each gene in the network circuit.**
State 1 represents the 0h timepoint, state 2 is 1 hour post infection, and state 3 is 3 hours post infection. Individual observations (dots) are connected by lines where multiple timepoints were measured. Blank spaces represent observations which either were not present in the experiment or could not be assigned to a discrete value with sufficient confidence to warrant inclusion as a constraint.

To assess the accuracy of these novel predicted interactions, we additionally measured the transcription of TNFAIP3 in L929 cells infected by 22–20 or 22–25 VSV via RT-PCR. While both viruses significantly upregulated TNFAIP3 transcription as predicted by the model (Fig 6), there was slightly significant statistical evidence that the magnitude of this upregulation was lower in cells infected by 22–25 (p<0.1)

## 4 Discussion

We constructed a mechanistic regulatory circuit model of the intracellular signaling pathways governing Type I IFN response to VSV infection based on causal relationships between the RIG-I and NFκB pathways documented in published scientific literature and experimental observations. The model was used to simulate observed responses in L929 cells using a binary (or "Boolean") logical framework. The initial model contained hypothetical connections between each VSV strain and every cellular entity in the network, such that each viral mutant could directly target any immune network molecule. By constraining this binary network model to support data obtained from *in vitro* experiments with different VSV strains, we identified a putative minimal set of targets necessary to reproduce the observed cellular response to infection by each VSV strain. The sets of targets shared by each pair of viruses suggest that the inability of 22–20 VSV to suppress host IFN production can be attributed to a failure to block NFκB activation.

After iteratively fixing or removing hypothetical viral actions based on the degree of consensus among candidate models best supporting the reference data, all but 4 of the initial 111

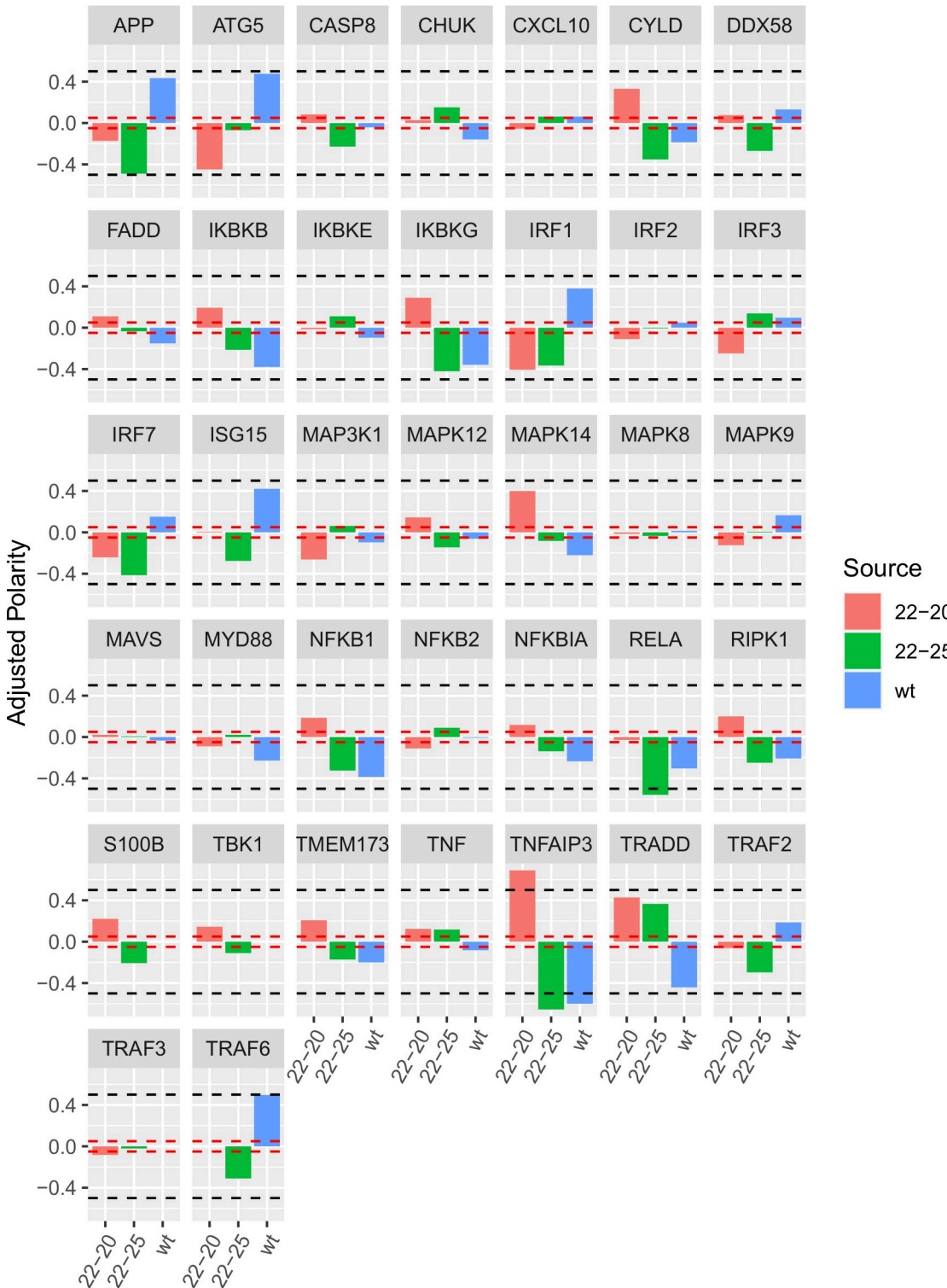

**Fig 5. Assignment of candidate edges after one generation of model parameterization.** Edges with adjusted polarity ≥0.5 or ≤-0.5 (black dashed lines) were fixed with their consensus polarity for subsequent model generations; edges with adjusted polarity between -0.05 and 0.05 (red dashed lines) were removed. This process was continued in an iterative fashion until candidate models converged on consensus values for edge inclusion and polarity (Table 1).

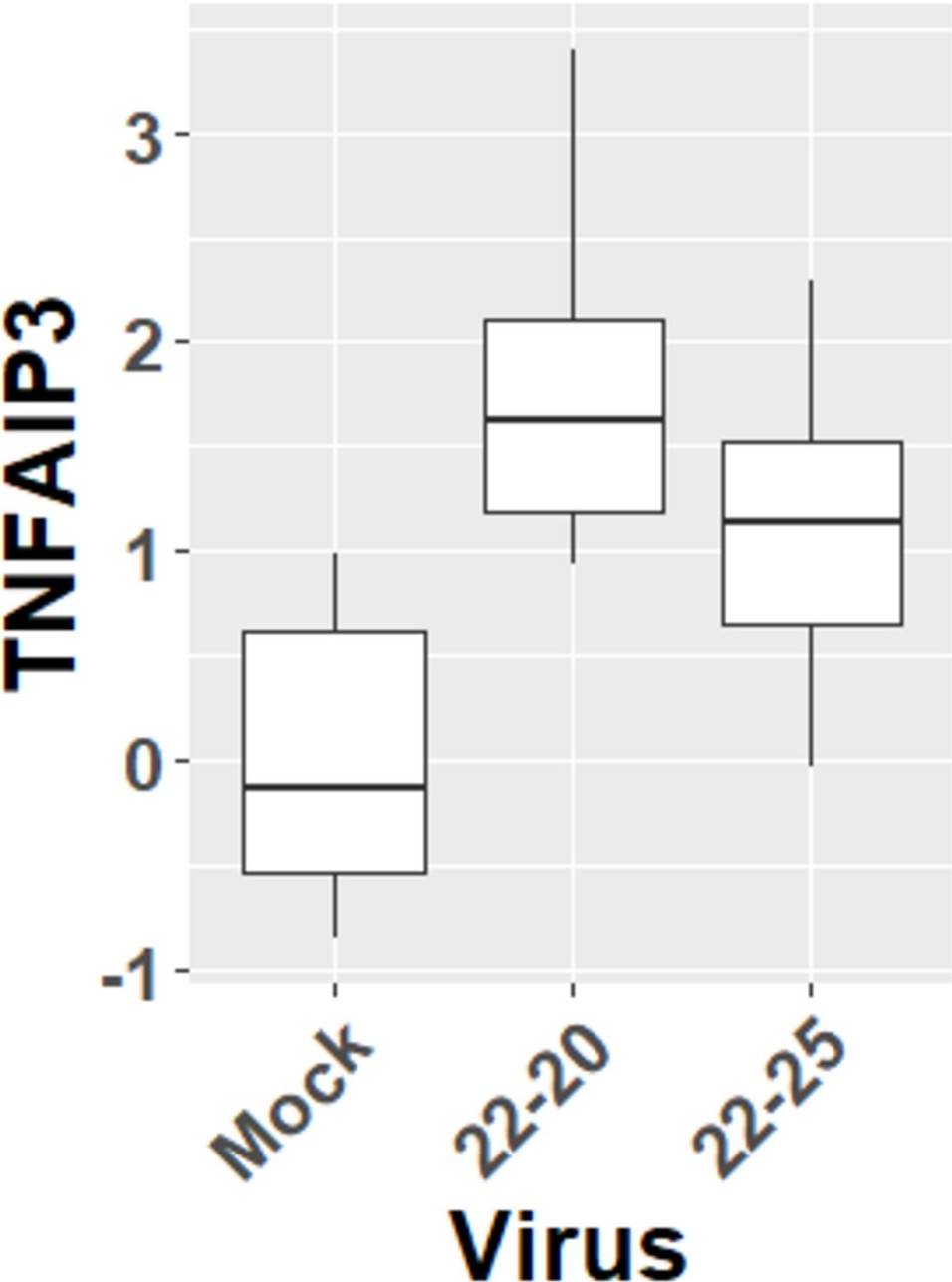

**Fig 6. Differential regulation of TNFAIP3 by 22–20 and 22–25 VSV.** The log2 fold change relative to uninfected (Mock) cells is displayed. TNFAIP3 mRNA is significantly upregulated by both 22–20 and 22–25 relative to Mock ($p < 0.05$), and the magnitude of upregulation is greater in 22–20 ($p < 0.1$).

could be confidently assigned. While different thresholds for edge fixation or removal could alter the outcome of this process, the applied criterion of agreement over a majority of models ensures that the final model is consistent with the best-performing candidate parameter sets. The mathematical problem posed is, by its nature, under-constrained, and the final hypothetical connectivity suggested by these models should be taken first as a suite of testable hypotheses in future experiments. It is also important to note that the host protein entities represented in the network circuit (Fig 2) are subject to other regulatory signals in addition to the viral

effects predicted by our modeling. For example, 22–25 VSV is predicted to exert an inhibitory effect on MAP3K1; however, MAP3K1 also receives activating signals from TRAF2 and TRAF6. These activating signals may counteract or overcome the inhibitory action of 22–25 VSV, such that it is possible for MAP3K1 activation to increase over time even during 22–25 VSV infection (Fig 3). The final set of predicted actions in Table 1 must be understood within the broader regulatory context of the network circuit in Fig 2.

There are striking similarities between the predicted actions of wt with 22–25. The targets for wt and 22–25 are functionally similar, with close connections to NFκB-dependent signaling. Predicted targets for 22–20, however, involve both NFκB and IRF family members. Wt VSV is known to have complex mechanisms of immune evasion, mainly dependent on the M protein [6, 9, 50]. The D52G mutation in the M protein of the 22–20 strain appears to have perturbed its ability to block NFκB activation in L929 cells. Thus, 22–20 VSV, unlike the wt and 22–25 strains, cannot rely on NFκB inhibition for immune evasion, and instead inhibits

**Table 1. Predicted targets for each VSV with polarity and inclusion fixed after 8 generations of parameterization.**

| Target | wt | 22–25 | 22–20 |
|---|---|---|---|
| IRF2 | | | negative |
| NFKBIA | | | negative |
| TRAF3 | | | negative |
| DDX58 | | negative | |
| ISG15 | | negative | |
| TBK1 | | negative | |
| CASP8 | | negative | positive |
| IKBKG | | negative | positive |
| S100B | | positive | |
| MAPK14 | | positive | positive |
| NFKB2 | | positive | positive |
| FADD | negative | | negative |
| IRF3 | negative | | negative |
| IKBKE | negative | negative | |
| RELA | negative | negative | |
| RIPK1 | negative | negative | |
| TNF | negative | negative | negative |
| TRAF2 | negative | negative | negative |
| TNFAIP3 | negative | negative | positive |
| CXCL10 | negative | positive | negative |
| IRF7 | positive | | negative |
| MAPK12 | positive | | negative |
| MYD88 | positive | | negative |
| APP | positive | | positive |
| CYLD | positive | | positive |
| MAPK9 | positive | | positive |
| MAP3K1 | positive | negative | |
| TRAF6 | positive | negative | |
| ATG5 | positive | positive | |
| CHUK | positive | positive | |
| NFKB1 | positive | positive | |
| IRF1 | positive | positive | negative |
| TRADD | positive | positive | positive |

host IFN production via an independent suppressor function [34]. These predictions can be partially validated by comparison with existing descriptions of VSV targets. VSV bearing a non-functional M protein has been shown to be a strong activator of TNFAIP3 [51], though wild-type VSV was not a subject of the experiment. Our model independently corroborates this result, and extends it to suggest that the property of TNFAIP3 activation may be specific to VSV strains bearing inactivating mutations in the M protein. The strain-specific regulatory interactions predicted by the computational modeling procedure are corroborated by the confirmatory RT-PCR we performed to measure TNFAIP3 transcription in infected cells. While both 22–20 and 22–25 strains upregulated TNFAIP3 relative to uninfected cells, resulting in their converging to similar values on a binary scale, we observed greater upregulation by 22–20 VSV, consistent with the differential regulation of TNFAIP3 by these strains predicted by our model.

While wt VSV has been observed to activate targets in addition to TBK1 [52], our model does not require both wt-M-bearing strains of VSV to interact with TBK1 directly. Since inhibition of IKKβ has been found to exacerbate inflammatory symptoms in sepsis [53], it is plausible that VSV strains bearing mutant M protein might favor early activation of NFκB-dependent signaling in infected cells. Conversely, the wt and 22–25 strains were predicted to activate MyD88, a critical component of IL-1R and TLR signaling pathways. In the context of TLR4, preferential engagement of MyD88-dependent signaling pathways appears to act in opposition to the TRIF-dependent pathway which governs IFN responses [54, 55]. A similar effect may manifest itself in the context of RIG-I/NFκB responses to VSV infection: by activating MyD88 but not IKK, wt VSV may direct the host inflammatory response away from early production of antiviral mediators. Pharmacological suppression of ADAM15, an inhibitor of TRIF activity and consequent IFN production, has been observed to increase inflammation in response to VSV infection in vitro [56]. The VSV-resistant PC-3 cell line showed reduced matrix metalloprotease activity and metastasis upon ADAM15 suppression (54), suggesting that this modulation of the inflammatory response by VSV may also hold true in cancer cells. In this study, we report novel predictions of potential causal mechanisms governing the dynamics of immune evasion by VSV. Based on RNA-Seq data and known regulatory interactions between components of the RIG-I and NFκB signaling pathways, we used constraint satisfaction programming to infer likely targets of direct molecular interactions by wt, 22–20, and 22–25 strains of VSV. Our simulations predict that VSV strains bearing the wt and M[D52G] forms of the M protein differentially regulate host inflammatory responses, in particular NFκB- and IRF-dependent signaling. These predictions would have been very time-consuming and costly to obtain by traditional experimental means: our constraint satisfaction approach has enabled us to leverage minimal observations to generate testable mechanistic hypotheses. In future work, we aim to extend this model by increasing its resolution beyond binary logic and testing these predicted interactions in vitro.

## Author Contributions

**Conceptualization:** Matthew C. Morris, Gordon Broderick, Maureen C. Ferran.

**Data curation:** Matthew C. Morris, Thomas M. Russell, Wesley K. Wong.

**Formal analysis:** Matthew C. Morris, Thomas M. Russell.

**Investigation:** Matthew C. Morris, Thomas M. Russell, Wesley K. Wong, Maureen C. Ferran.

**Methodology:** Matthew C. Morris, Thomas M. Russell, Cole A. Lyman, Wesley K. Wong, Maureen C. Ferran.

**Project administration:** Matthew C. Morris, Maureen C. Ferran.

**Resources:** Gordon Broderick, Maureen C. Ferran.

**Software:** Matthew C. Morris, Cole A. Lyman.

**Supervision:** Gordon Broderick, Maureen C. Ferran.

**Writing – original draft:** Matthew C. Morris.

**Writing – review & editing:** Cole A. Lyman, Gordon Broderick, Maureen C. Ferran.

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
