## [Decision Letter · Decision Letter 0]

5 Aug 2021

PONE-D-21-19939

Computational prediction of intracellular targets of wild-type or mutant vesicular stomatitis matrix protein

PLOS ONE

Dear Dr. Ferran,

Thank you for submitting your manuscript to PLOS ONE. After careful consideration, we feel that it has merit but does not fully meet PLOS ONE’s publication criteria as it currently stands. Therefore, we invite you to submit a revised version of the manuscript that addresses the points raised during the review process.

In particular, please pay close attention to the comments of the reviewer. As Editor, I also request that you provide some experimental validation of your model's predictions that 22-25 VSV has direct inhibitory actions on key elements of the NFκB signaling pathway, while 22-20 fails to inhibit this pathway.

We look forward to receiving your revised manuscript.

Kind regards,

Stephen J Polyak

Academic Editor

PLOS ONE

Journal Requirements:

“MCF

R15CA246419

National Cancer Institute (National Institutes of Health)

https://www.cancer.gov/

No role in the above.”

3. Please ensure that you refer to Figure 5 in your text as, if accepted, production will need this reference to link the reader to the figure.

Reviewers' comments:

Reviewer's Responses to Questions

**Comments to the Author**

1. Is the manuscript technically sound, and do the data support the conclusions?

Reviewer #1: Partly

2. Has the statistical analysis been performed appropriately and rigorously? 

Reviewer #1: N/A

3. Have the authors made all data underlying the findings in their manuscript fully available?

Reviewer #1: No

4. Is the manuscript presented in an intelligible fashion and written in standard English?

Reviewer #1: Yes

5. Review Comments to the Author

Reviewer #1: The article by Morris et.al implements Boolean modelling to predict the targets of vesicular stomatitis matrix protein to understand the role in suppression of type-1 interferon pathway.

While the concept of implementing the logical model is reasonable, the paper lacks description. Most concepts are not described sufficiently well in the manuscript. It is important certain concepts/ model techniques are explicitly described rather than just citing papers that describe those techniques so that the article is easy to follow for even non-computational readers.

The authors perform differential expression analysis (section 2.2 ), the results for the same are not been presented.

Section 2.3 Pathway studio is used to perform text mining, however the search key word and criteria used in describing the interactions is not clearly described. It will be useful to have this information for reproducibility using open-source software as well.

Section 3.1 The discretization method doesn’t explain how the cutoff was set. Mathematical expression or textual interaction map is missing. Only a figure is provided (Figure-2)

Section 3.3 Biological relevance/importance of Edge fixation is not detailed.

Better description of PCA analysis is required.

The authors have performed a sizable amount of analysis but only a fraction of it has been presented in the results. Also, few parts (ex:PCA analysis) are not even described clearly.

6. PLOS authors have the option to publish the peer review history of their article (what does this mean?). If published, this will include your full peer review and any attached files.

Reviewer #1: No

---

## [Author Response · Author response to Decision Letter 0]

22 Dec 2021

Please see our response to reviewer document

---

## [Decision Letter · Decision Letter 1]

12 Jan 2022

Computational prediction of intracellular targets of wild-type or mutant vesicular stomatitis matrix protein

PONE-D-21-19939R1

Dear Dr. Ferran,

We’re pleased to inform you that your manuscript has been judged scientifically suitable for publication and will be formally accepted for publication once it meets all outstanding technical requirements.

Kind regards,

Stephen J Polyak

Academic Editor

PLOS ONE

Additional Editor Comments (optional):

Reviewers' comments:

Reviewer's Responses to Questions

**Comments to the Author**

1. If the authors have adequately addressed your comments raised in a previous round of review and you feel that this manuscript is now acceptable for publication, you may indicate that here to bypass the “Comments to the Author” section, enter your conflict of interest statement in the “Confidential to Editor” section, and submit your "Accept" recommendation.

Reviewer #1: All comments have been addressed

2. Is the manuscript technically sound, and do the data support the conclusions?

Reviewer #1: Yes

3. Has the statistical analysis been performed appropriately and rigorously? 

Reviewer #1: Yes

4. Have the authors made all data underlying the findings in their manuscript fully available?

Reviewer #1: Yes

5. Is the manuscript presented in an intelligible fashion and written in standard English?

Reviewer #1: Yes

6. Review Comments to the Author

Reviewer #1: It would be good to add a graphical abstract for clarity. All the questions are addressed in the manuscript as well.

7. PLOS authors have the option to publish the peer review history of their article (what does this mean?). If published, this will include your full peer review and any attached files.

Reviewer #1: No

---

## [Editor Report · Acceptance letter]

20 Jan 2022

PONE-D-21-19939R1 

Computational prediction of intracellular targets of wild-type or mutant vesicular stomatitis matrix protein 

Dear Dr. Ferran:

I'm pleased to inform you that your manuscript has been deemed suitable for publication in PLOS ONE. Congratulations! Your manuscript is now with our production department. 

Kind regards, 

on behalf of

Dr. Stephen J Polyak 

Academic Editor

PLOS ONE